



# The Origins of a Near-Ecliptic Merged Interaction Region as a Magnetic-Cloud like Structure Embedded in a Co-rotating Interaction Region

Megan L. Maunder[1], Claire Foullon[1], Robert Forsyth[2], David Barnes[3], and Jackie Davies[3]

[1]University of Exeter, Department of Mathematics and Statistics, Exeter EX4 4QF, UK
[2]Department of Physics, Imperial College London, London SW7 2AZ, UK
[3]STFC RAL Space, Rutherford Appleton Laboratory, Harwell Campus, Oxfordshire OX11 0QX, UK

**Correspondence:** Claire Foullon (c.foullon@exeter.ac.uk)

**Abstract.**

Using remote-sensing and in-situ observations across multiple spacecraft with complimentary methods of analysis, we investigate a Magnetic Cloud Like-structure (MCL) observed in-situ on 3-4 July 2007 near the ecliptic at OMNI, STEREO-A and -B (all within 15° longitude of Earth). The MCL is entrained in a Corotating Interaction Region (CIR) originating in the Northern

heliospheric sector, to create a Merged Interaction Region (MIR). This event allows the comparison of MIR observations at different longitudes showing differences in size, formation of sheath, presence of forward and reverse waves and small-scale structuring, demonstrating the progression of the interaction between the CIR and MCL from West to East. In order to explore its origins further, we compare the MIR with the (Interplanetary) Coronal Mass Ejection (ICME/CME) studied in Maunder et al. (2022) in the mid-latitudes at *Ulysses* containing a Magnetic Cloud (MC) and present a comprehensive discussion of

the challenges posed by observing and relating transients not in alignment, across different latitudes and longitudes, and in different solar wind environments. As the CME propagates almost directly towards *Ulysses*, we find through fitting and modelling that its flanks could also potentially skim the near-ecliptic spacecraft. Length-scale analysis appears to be consistent with this configuration. However, local expansion velocities of the MCL/MC indicate compression near the ecliptic and expansion at *Ulysses* and the magnetic flux rope orientations and helicities at the different latitudes oppose each other. The CIR likely

causes more compression and re-aligns the transient axis orientation near the ecliptic while a High Speed Stream (HSS) from the Southern sector propagates directly into the back of the ICME/MC near the mid-latitude. Opposing signs of helicity could provide indications of flux added in the first stages of CME evolution or magnetic reconnection with the Heliospheric Current Sheet (HCS). These observations and analyses demonstrate the continued challenge of modelling and fitting the propagation of transients embedded in complex solar wind environments. We note some of the caveats and limitations in the methods and

highlight the use of multi-spacecraft analysis to disentangle the origin and formation of ICME substructures from the solar wind and other transients.



# 1 Introduction

The heliosphere is formed by the continuous outflow of the solar wind (Parker, 1958), in which we observe both small and large scale transients. Coronal Mass Ejections (CMEs) and their heliospheric counterparts (Interplanetary CMEs, ICMEs), are large-scale transients known to drive interplanetary shocks (Gosling et al., 1975). ICMEs with Magnetic Clouds (MCs) are a subset of ICMEs with distinct magnetic field characteristics: an enhanced magnetic field and smooth, large-scale rotation of the magnetic field vectors (Burlaga, 1984, 2001). Marubashi (1986) and Bothmer and Schwenn (1998) suggest that a twisted magnetic flux tube can approximate the global magnetic structure of a MC, and fitting models to the magnetic field time series may provide an estimate of its large-scale topology (Lepping et al., 1990; Bothmer and Schwenn, 1998). A MC is usually identified when a spacecraft passes near its cloud axis. Magnetic Cloud Like-structures (MCLs) are magnetic structures identified using the same definition as a MC, except that a traditional flux-rope model cannot be fitted (Lepping et al., 2005; Foullon et al., 2007). Wu et al. (2006) found that the occurrence rate of MCLs and joint sets (MCs plus MCLs) are related to both the solar activity and the CME occurrence rate. This correlation is not observed with MCs alone. It is possible that all ICMEs contain flux-ropes but that some are not recognised as MCs only because the spacecraft trajectories skim the flanks (Marubashi, 1997; Kilpua et al., 2012, 2017).

Identifying ICMEs from in-situ data is a challenging and somewhat ambiguous task; there are around two dozen recognised signatures (Zurbuchen and Richardson, 2006), each with its own limitations (Richardson, 2014), and longitudinal separations of a few degrees between spacecraft can still result in significantly different observations and event properties (Davies et al., 2020). Single spacecraft in-situ observations enable limited analysis as they are only able to measure a single track through the ICME as it passes over the spacecraft, leading to discrepancies or incomplete knowledge of the instantaneous structure and composition of ICMEs (Russell and Mulligan, 2002; Howard, 2011; DiBraccio et al., 2015). According to Kilpua et al. (2012), 'ICME-like' transients are defined as lasting more than three hours but not meeting the maximum magnetic field criteria of $(B_{max}) > 7$ nT and duration of more than 10 hours to be classed as an ICME. The identification becomes even more challenging in the case of flux-ropes in complex solar wind environments (e.g. Rouillard et al., 2009b; Winslow et al., 2021). In such cases, multi-spacecraft observations become key to disentangling features to give a more accurate estimation of flux-rope properties and structure of ICMEs at different points, permitting the development of a more accurate implied 3D structure (e.g. Foullon et al., 2007; Möstl et al., 2009; Palmerio et al., 2019).

Stream Interaction Regions (SIRs), sometimes referred to as Corotating Interaction Regions (CIRs), are regions in which the ambient plasma is compressed (Pizzo and Gosling, 1994; Richardson, 2018) as a result of fast wind emerging behind slow wind from adjacent longitudinal sources interacting as they are brought into radial alignment by solar rotation. They are common high-density structures observed in situ during solar minimum (Gazis, 1996). Since coronal holes, the source of the fast wind at the Sun, tend to be long-lived, CIRs are the subset of SIRs observed at regular intervals of approximately the solar rotation period. Conventionally SIRs and CIRs are associated with the streamer belt and its embedded Heliospheric Current Sheet (HCS), which can extend north and south of the solar equator to heliographic latitudes of about $\pm 30\,^\circ$. For clarity, we refer to a CIR when observed near the equatorial region and to a SIR more broadly to include an interaction region resulting from





**Table 1.** Spacecraft positions on 4 July 2007.

| Spacecraft | Latitude (°) [HEE] | Longitude (°) [HEE] | Radial Distance (AU) |
|---|---|---|---|
| STEREO-A | 0.12 | 9.39 | 0.96 |
| OMNI (Earth) | 0.00 | 0.00 | 1 |
| STEREO-B | -0.26 | -6.39 | 1.08 |
| *Ulysses* | -32.85 | 49.06 | 1.48 |

a high latitude High Speed Stream (HSS). Merged Interaction Regions (MIRs) are complex structures arising from flux-rope interactions including, but not limited to, interactions with a HSS, SIR (Rouillard et al., 2009b, 2010b) or the HCS.

In the present study we investigate complex structures observed in-situ on 3-4 July 2007. Using remote-sensing and in-situ observations across multiple spacecraft near the ecliptic with complimentary methods of analysis, we demonstrate the presence
of a MCL embedded in a CIR, resulting in a MIR. We address (1) the presence of a CIR and how this combines with the MCL and is observed as a MIR, (2) the arrival times of the entrained MCL, which are consistent with an ICME entrained in a CIR, (3) the difference in the MIR observed at different longitudes with a small separation angle.

We include data sets from the STEREO (STEREO-A and STEREO-B, Kaiser et al., 2008) Sun Earth Connection Coronal and Heliospheric Investigation (SECCHI) HI-1 and HI-2 cameras, with respective elongation ranges of $4 - 24\,^\circ$ and $18 - 88\,^\circ$ in
the ecliptic plane (Eyles et al., 2009), the Plasma and Suprathermal Ion Composition Magnetometer (IMPACT Luhmann et al., 2008), the IMPACT Solar Wind Plasma Electron Analyzer (SWEA Luhmann et al., 2008), and the Plasma and Suprathermal Ion Composition (PLASTIC) investigation (Galvin et al., 2008). We also use in-situ observations from the OMNI virtual observatory, which combines multiple near-Earth (L1) data sets, including the Advanced Composition Explorer (ACE) (Stone et al., 1998) and Wind (Lepping et al., 1995); see King and Papitashvili (2005).
The spacecraft positions on 4 July 2007 are given in Table 1 in Heliographic Earth Ecliptic coordinate system (HEE) coordinates, and are shown in Fig. 1 as seen from above the North pole (panel (a)) and from 90° longitude to the East of the Sun-Earth line (panel (b)). On a heliospheric scale we can consider OMNI's position as Earth (rather than the adjusted Earth bowshock nose location). This event was observed very early into the STEREO mission, where both spacecraft are close to Earth; STEREO-A and STEREO-B have a longitudinal separation of $9.4\,^\circ$ and $5.8\,^\circ$ [HEE] from Earth respectively. Given their
HEE latitudes, we will refer to the STEREO-A, OMNI, and STEREO-B in-situ observations as 'near-ecliptic'.

Here our study focuses on the near-ecliptic in-situ observations, and we compare them with the contemporaneous solar event presented in Maunder et al. (2022) to investigate whether these events could be related. In Maunder et al. (2022), the multi-spacecraft study of a mid-latitude CME revealed by a unique orbital configuration permitted the analysis of remote-sensing observations from the twin STEREO -A and -B spacecraft and of its subsequent in situ counterpart outside the ecliptic plane, at
*Ulysses*. A triangulation method was applied to the STEREO/SECCHI COR2 coronagraph images of the CME, and a single-spacecraft geometrical fitting method to the time-elongation profile of the CME's apex from STEREO/SECCHI HI-B data. The





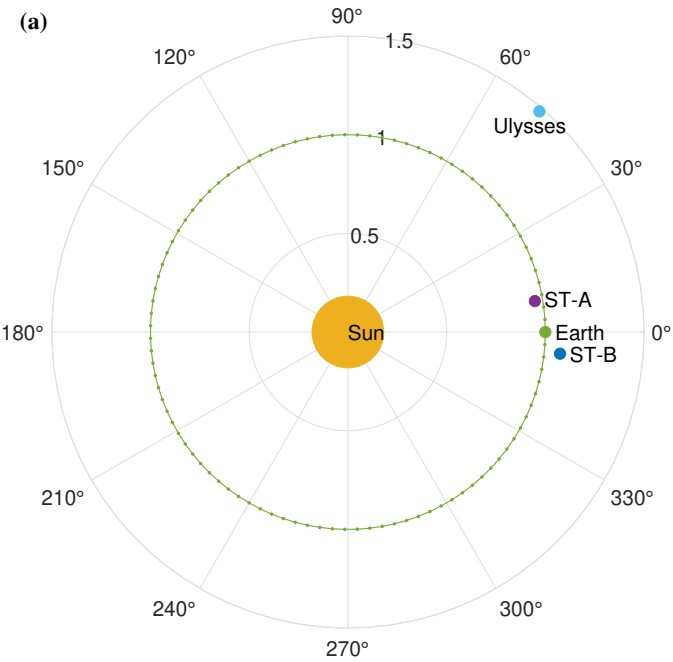

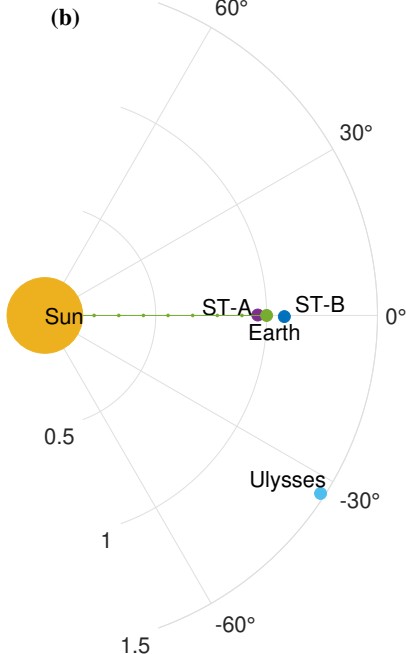

**Figure 1.** Spacecraft positions on 04 July 2007, (a) view from above the solar north pole including Earth's orbit (green dashed line). (b): view from 90° longitude to the East of the Sun-Earth line.

ICME is observed on 04 July 2007 by *Ulysses* near 1.5 AU and 33° latitude and 49° longitude [HEE]. This includes a preceding forward shock, followed by a sheath region, a MC, followed by a compression region driving a second forward-shock, due to a succeeding HSS interacting with the ICME. In particular, the MC had a duration of 24h, a West-North-East configuration, a

length scale of $\approx0.2$ AU, and mean expansion speed of $14.2$ $\mathrm{km\,s^{-1}}$ inferred through a Minimum Variance Analysis (MVA) and a length-scale analysis at *Ulysses*. The relatively small size of this ICME is likely to be a result of its interaction with the succeeding HSS which in itself drives the second forward shock into the back of the MC.

We present: the in-situ observations of ICME or ICME-like transients at STEREO-A, STEREO-B, and OMNI showing evidence for a MIR (Sect. 2); further analysis of the transients and a comparison of the observations against that of the event

observed at *Ulysses* presented in Maunder et al. (2022) (Sect. 3); followed by a summary of our findings (Summary).

## 2 Near-ecliptic observations

### 2.1 ICME and ICME-like transients

Figures 2, 3, and 4 show the structures observed at OMNI, STEREO-A, and STEREO-B respectively. Further observations from ACE are shown in Fig. 5 to complement those at OMNI (as the ACE data are not shown in the same time frame, similar





**Table 2.** MIR and MCL (STEREO/OMNI) / MC (*Ulysses*) Properties: the arrival time of the wave front $t_{FW1}$, flux-rope leading edge $t_{MCL}$ or $t_{MC}$, second forward wave $t_{FW2}$, trailing reverse waves $t_{RW}$, the mean velocity, $\langle V \rangle$, velocity $V_{MCL}$ or $V_{MC}$, leading edge velocity, maximum total magnetic field strength $B_{max}$, average magnetic field $\langle B \rangle$.

| Spacecraft | $t_{FW1}$ | $t_{MC(L)}$ | $t_{FW2}$ | $t_{RW}$ | $V_{MC(L)}$ | $B_{max}$ | $\langle B \rangle$ |
|---|---|---|---|---|---|---|---|
| STEREO-A | 03 Jul 07:40 | 03 Jul 16:16 | 04 Jul 07:00 | 04 Jul 22:00 | 350 | 13 | 7 |
| OMNI | (03 Jul 11:00) | 03 Jul 11:00 | 03 Jul 19:00 | 04 Jul 08:10 | 346 | 11 | 9 |
| STEREO-B | (03 Jul 04:00) | 03 Jul 04:00 | 03 Jul 14:10 | 04 Jul 08:00 | 360 | 10 | 7 |
| *Ulysses* | 04 July 08:00 | 04 July 13:00 | 05 July 13 | 05 July 20:50 | 396 | 13.3 | 11.5 |

observations to those in Fig. 2 are shown for context). The MCL boundaries are indicated by magenta vertical lines and specific MCL properties are given in Table 2. A CIR is also observed across all spacecraft and Table 2 gives the arrival time of each part of the transient and CIR combined, which we refer to as the MIR. The combined ICME-CIR structure will be discussed as an MIR in Sect. 2.3.

Panels (a)-(d) show magnitude and normalised components of the magnetic field vector. For the twin STEREO observations,
they are presented in the Heliographic Radial Tangential Normal (RTN) system of reference (Burlaga, 1984) and normalised. In this coordinate system, $\boldsymbol{R}$ points from the Sun to the spacecraft, $\boldsymbol{T}$ is the Sun's rotation vector crossed into $\boldsymbol{R}$ (thus towards west for a spacecraft near the ecliptic) and $\boldsymbol{N}$ completes the right-handed system. The magnetic field components of the OMNI observations are given in the Geocentric Solar Ecliptic (GSE) system of reference. In the OMNI data, the correspondence between GSE and RTN is then $\boldsymbol{B_X} \equiv -\boldsymbol{B_R}$, $\boldsymbol{B_Y} \equiv -\boldsymbol{B_T}$, $\boldsymbol{B_Z} \equiv \boldsymbol{B_N}$ for the magnetic field components and so we refer
to the transformed components for ease of comparison. The MCL is identified by an enhanced magnetic field followed by a clear rotation in the $\boldsymbol{B_T}$ and $\boldsymbol{B_N}$ vectors in STEREO-A and -B and in the $-\boldsymbol{B_Z}$ vector in OMNI. The flux rope leading edge (labelled 'MCL') is first observed at STEREO-B at 04:00 UT on 3 July, then at OMNI at 11:00 UT and at STEREO-A at 16:16 UT the same day (time $t_{MCL}$). The MCL, identified between time $t_{MCL}$ and the trailing edge at time $t_{FW2}$, has a mean magnetic field intensity (averaged over the time range between $t_{MCL}$ and $t_{FW2}$) $\langle B \rangle = 7, 9, 7$ nT in STEREO-A, OMNI,
and STEREO-B respectively. The structures at STEREO-A and STEREO-B meet the ICME criteria proposed by Kilpua et al. (2012); only the MCL structure at OMNI lasts less than 10 hours and can therefore be classified as an ICME-like structure.

In panel (e), the MCL is characterised by a consistent and relatively slow proton speed (e.g. Lepping et al., 1990) with a mean solar wind speed (and standard deviation) of 337 ($\pm$ 6.8), 362 ($\pm$ 14.1), 411 ($\pm$ 32.9) km s$^{-1}$ at STEREO-A, OMNI, and STEREO-B respectively (again averaged over the time range between $t_{MCL}$ and $t_{FW2}$). The proton temperature, in panel
(f) is below the expected proton temperature for STEREO-A and OMNI, indicative of a MC, but matches approximately the expected solar wind temperature T$_{ex}$ (over plotted in blue, Richardson and Cane, 1995; Lopez, 1987) across all three near-ecliptic spacecraft. Panel (g) shows a drop in proton density. Panel (h) shows the total pressure (black), magnetic pressure (purple) and proton pressure (navy); here we observe a drop in proton pressure, consistent with the passage of a MC (Gosling, 1994; Gosling et al., 1994).





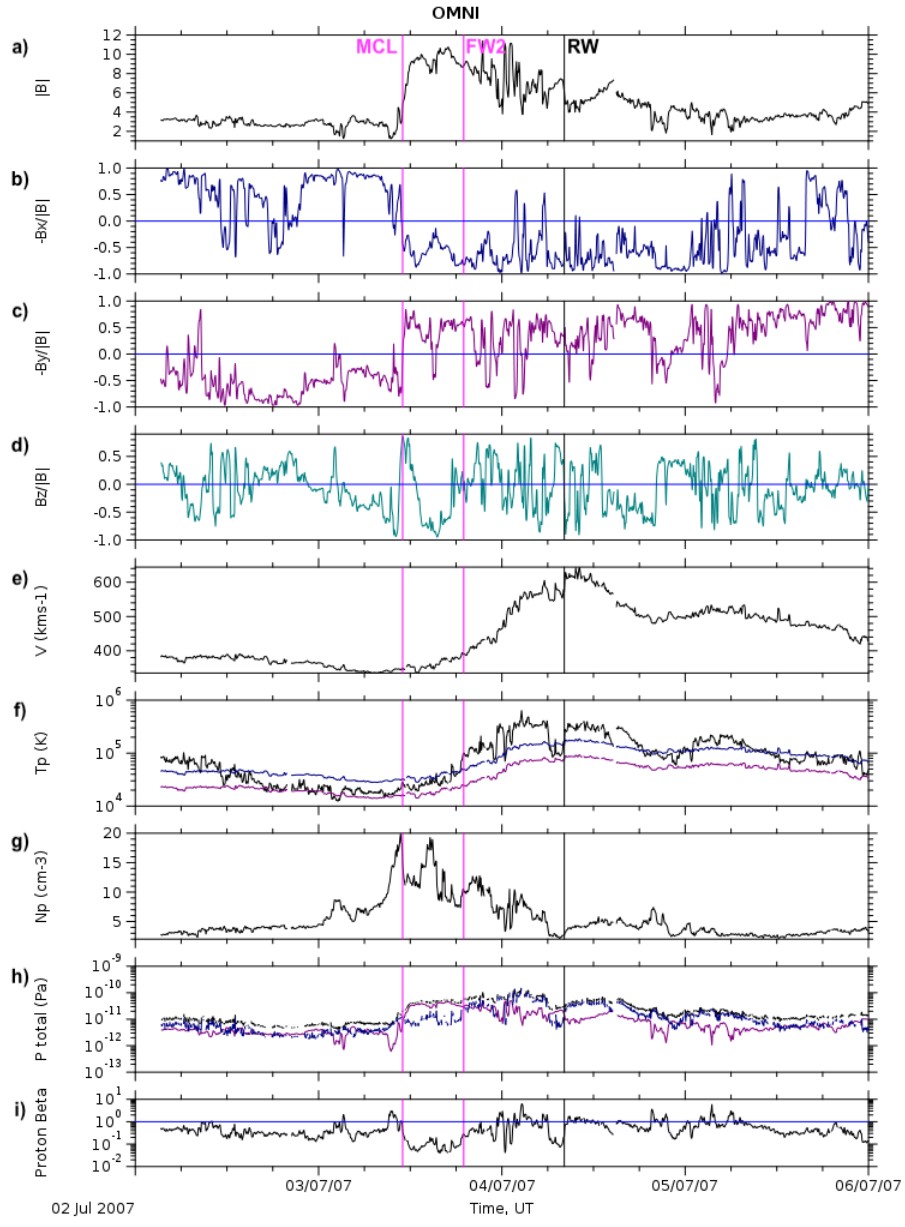

**Figure 2.** The MIR observed on 3 - 4 July 2007 by OMNI. (a) Magnitude and (b-d) normalised components of the magnetic field vector; (e) proton speed; (f) proton temperature (black), expected temperature calculated from the observed solar wind speed (blue); half the expected proton temperature, (purple); (g) proton density; (h) total pressure (black), magnetic pressure (purple) and proton pressure (navy); (i) proton plasma beta with a threshold of 1 (blue). Magenta vertical lines labelled 'MCL' and 'FW2' represent the leading and trailing edges of the MCL, respectively; the black vertical line labelled 'RW' represent the reverse pressure wave behind the compression region.



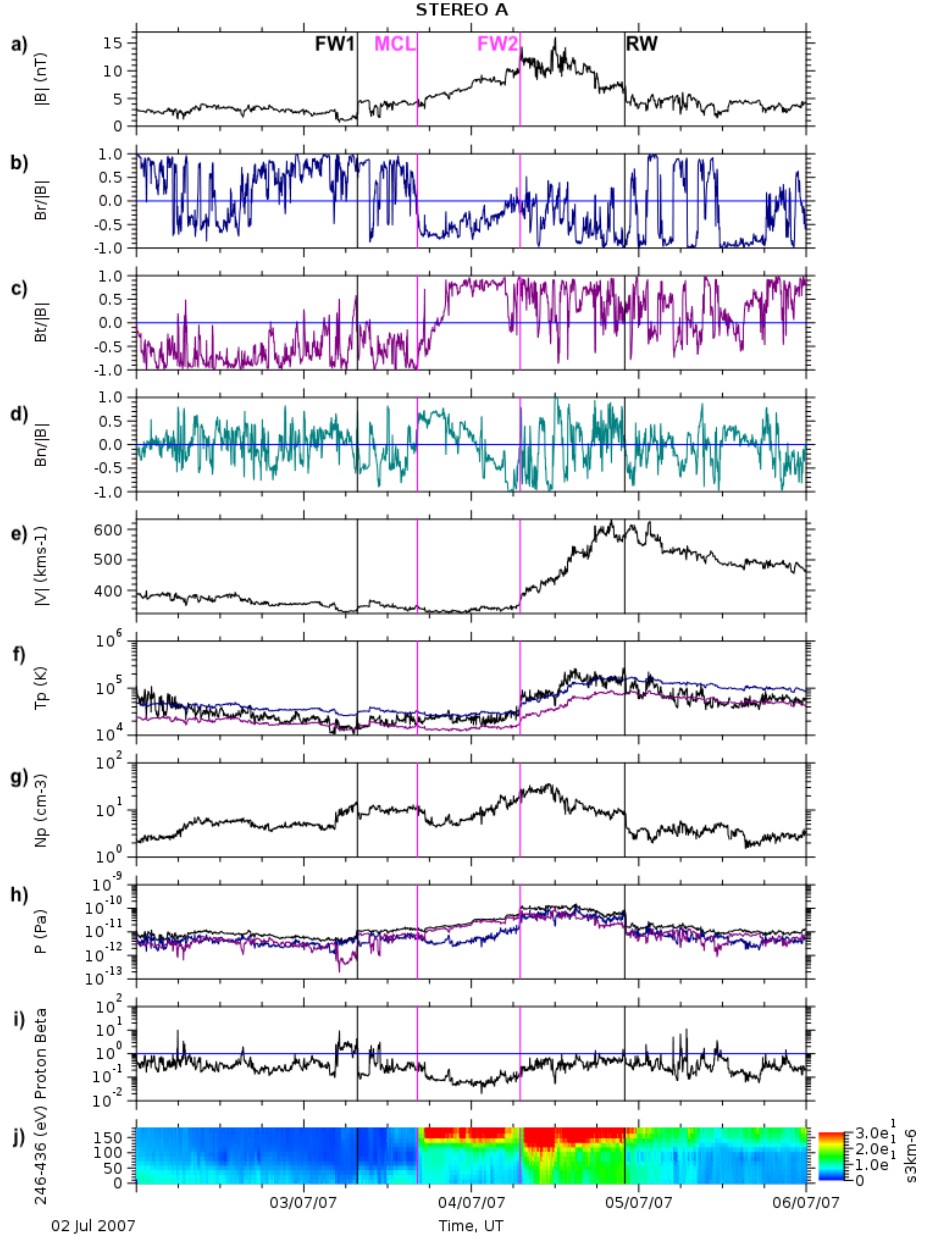

**Figure 3.** The MIR observed on 3 July 2007 at STEREO-A. Captions for panels and labels are as those in Fig. 2 with the addition of (j) the electron pitch angles at 272 eV and 'FW1' the black vertical line representing the pressure wave in front of the sheath.

The proton-beta, $\beta_p$ in panel (i), is consistently less than 1 across the MCL (0.01-0.38, 0.04-0.27, 0.03-0.67 in STEREO-A, OMNI, STEREO-B respectively) and we see a distinct drop in comparison with the surrounding plasma. This is a convincing





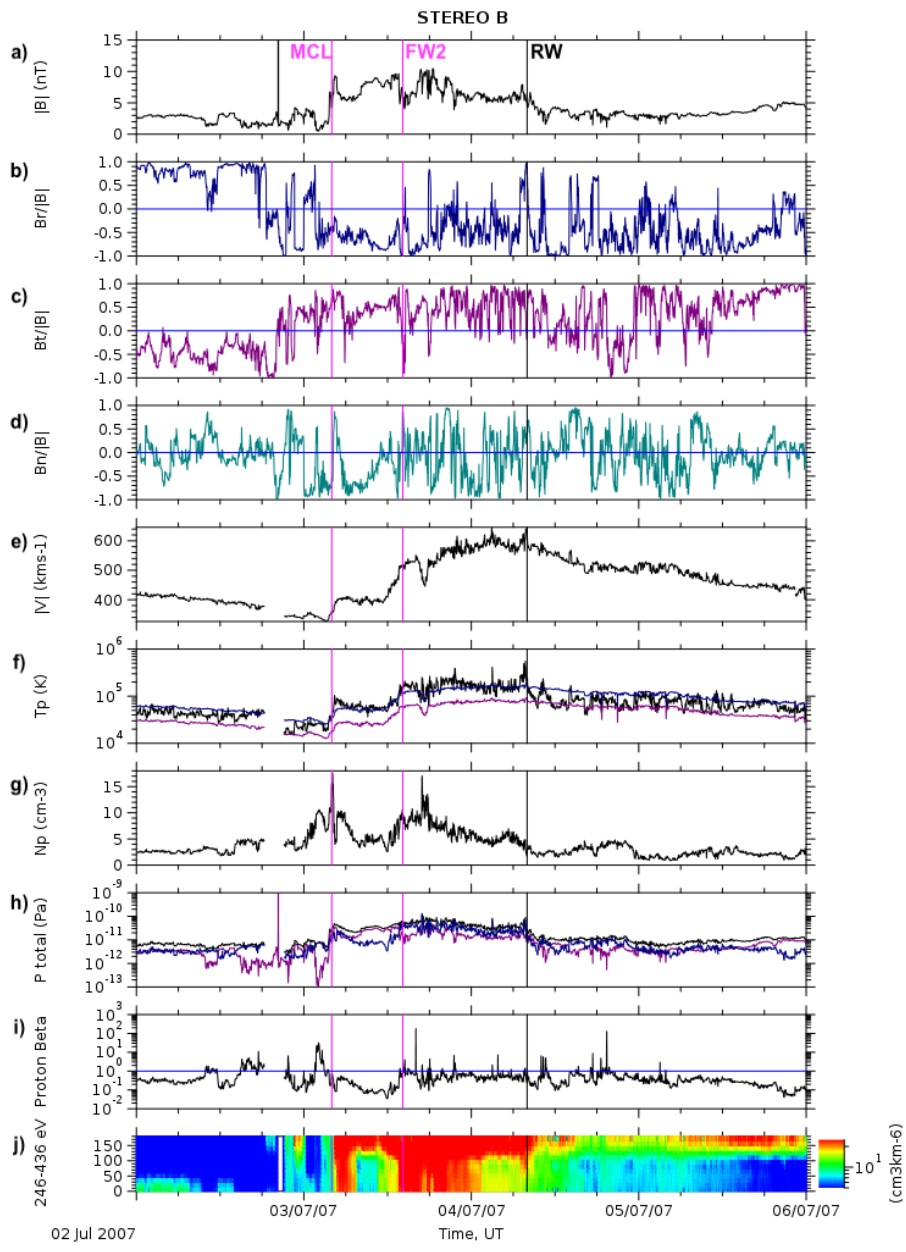

**Figure 4.** The MIR observed on 3 - 4 July 2007 by STEREO-B. Captions for panels and labels are as those in Fig. 3.

signature at all three spacecraft. For STEREO-A and -B we present the suprathermal electron pitch angle distribution, in panel (j). The ACE electron pitch angle data is shown in panel (g) of Fig. 5 (OMNI does not include pitch angle data). This is predominately anti-sunward in STEREO-A and ACE with some spreading in STEREO-B.



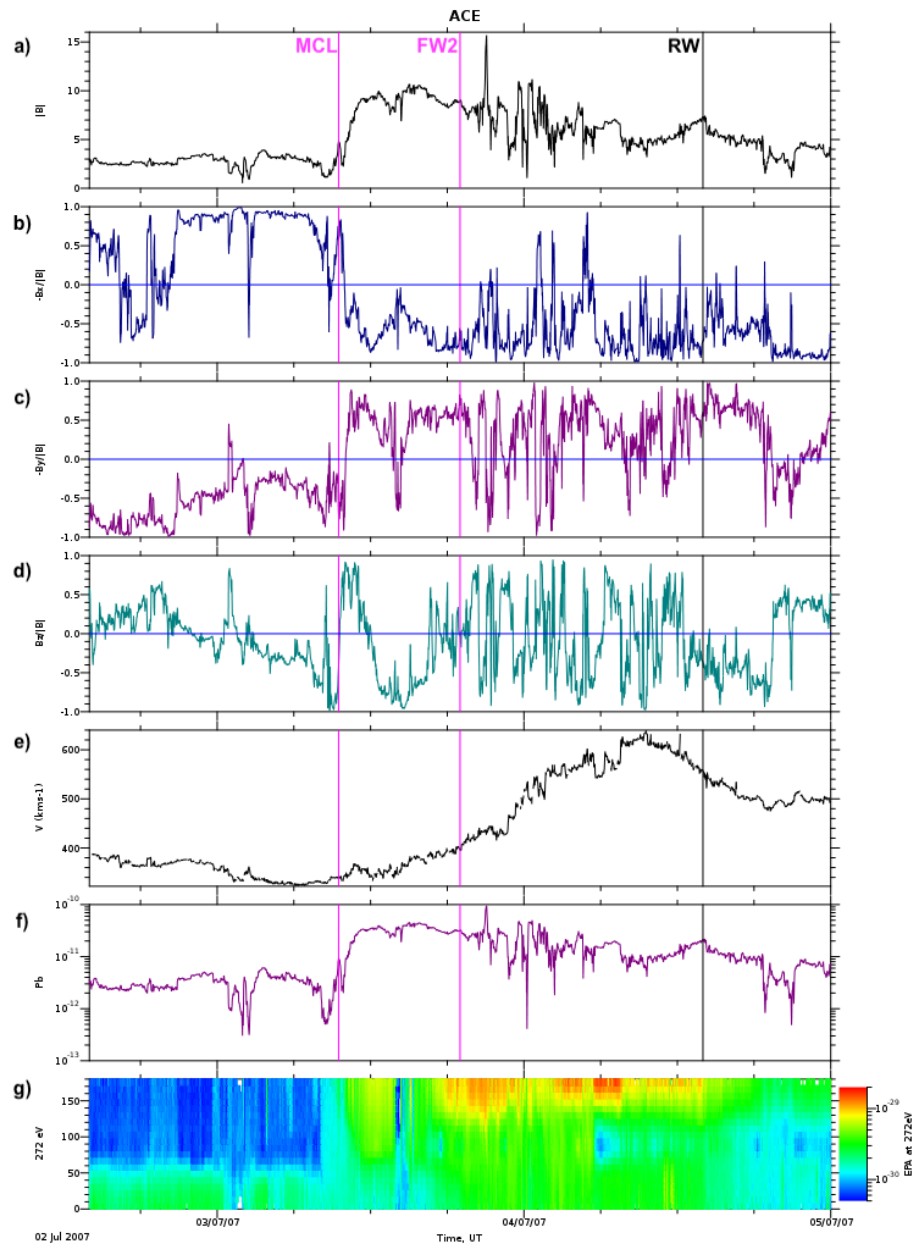

**Figure 5.** The MIR observed on 3 - 4 July 2007 by ACE. Captions for panels (a)-(f) are as in Fig. 2, panel (g) as in panel (j) in Fig. 3.

At STEREO-A, we observe a preceding forward pressure wave (FW1) at 07:40 UT on 3 July, characterised by a pressure and speed increases. This wave, 8:36 hours before and a result of plasma build up at the leading edge of the MCL, is not evident in the other near-ecliptic spacecraft observations. Instead, if a pressure wave is present at OMNI and STEREO-B, it



**Table 3.** Magnetic sector and electron pitch angle characteristics for each part of the transient and flux-rope orientations for the MC(L) at each spacecraft.

| | Magnetic Sector and Electron Pitch Angle Characteristics | | | Flux-rope Orientation |
| --- | --- | --- | --- | --- |
| | Before | MCL/MC | After | |
| STEREO-A | Away, $0°$ | Towards, $180°$ | Towards $180°$ | North-West-South (NWS) |
| ACE (OMNI) | Away, $0°$ | Towards, $180°$ | Towards, $180°$ | North-West-South (NWS) |
| STEREO-B | Away, $0°$ | Towards, $180°$ | Towards, $180°$ | North-West-South (NWS) |
| | with brief reversals, $0 - 180°$ | (with spreading) | (with spreading) | |
| *Ulysses* | Away, $0°$ | (Counter-Streaming) | (Counter-Streaming) | West-North-East (WNE) |

coincides with the start of the MCL (this time is shown between brackets in Table 2). Similarly, the trailing edge of the MCL at STEREO-A coincides with a second forward wave (FW2); this is less evident at the other spacecraft. At all near-ecliptic

spacecraft, a CIR achieving speeds above $600 \mathrm{~km\,s}^{-1}$ runs into the back of the MCL. At STEREO-A, this interaction drives the forward wave (FW2) at the front of the compression region, the latter being the CIR. At all spacecraft, a reverse wave (RW) propagates back into the fast wind behind the CIR, marking the rear of the CIR.

Thus FW1 and FW2 are reasonably clear at STEREO-A but not obviously present in STEREO-B and OMNI data. The front sheath (FS) and the MCL at STEREO-A form an ICME. The ICME at STEREO-B and the ICME-like transient at OMNI

contain only the MCL. The MCL is followed by two distinct boundaries, FW2 and RW: at STEREO-A, they are the forward and reverse waves from a CIR, while at STEREO-B and OMNI, FW2 only marks a boundary between the MCL and CIR and RW marks the location of a reverse wave. Comparing the speed profile at STEREO-B with those at the other spacecraft, it is possible that the wave driven by the CIR compression has propagated through the front of the MCL. In other words, the MCL is more entrained at STEREO-B than at the other spacecraft. The reverse wave RW is detected at 08:00 UT, 08:10 UT and 22:00

UT 04 July, in STEREO-B, OMNI and STEREO-A, respectively 17:50, 13:10, 15:00 hours later than the FW2 boundary.

## 2.2 Magnetic orientation

In Figs. 3-4, between the leading and trailing edges of the MCL, we distinguish a clear coherent rotation of the magnetic field vector with a low level of magnetic field-magnitude fluctuations, and low values of proton plasma beta. Although a MVA analysis did not reveal reliable axis orientations, these are clear signatures of MCL and support the flux-rope interpretation. We

adopt the method proposed by Bothmer and Schwenn (1998) and Mulligan et al. (1998) to deduce the magnetic configuration of the MCLs, and the sign of helicity, a measure that describes how the magnetic field lines are wound around each other (applied in the same manner as Dasso et al., 2006; Foullon et al., 2007; Kilpua et al., 2017). The method assumes that a MC (or MCL as in this case) possesses a specific magnetic configuration, of which the sense of the rotation indicates the sign of its helicity. We also use the suprathermal electron pitch angle characteristics from STEREO-A, -B, and ACE (see Fig. 3, 4 and 5)





to assist in the interpretation of the magnetic topology of the field lines local to the spacecraft. The flux-rope orientation and characteristics are listed in Table 3.

At STEREO-A, the local orientation of the MCL normalised magnetic field vector, as shown in panels (c,d) of Fig. 3, turns from North ($B_N > 0$), to West ($B_T > 0$), to South ($B_N < 0$) on the cloud's axis. This flux-rope cloud can be interpreted as a North-West-South (NWS) cloud with left-handed (negative) helicity. In panel (c) the (normalised) $B_R$ component of the

magnetic field is mostly negative during and after the MCL. In panel (j) we find the streaming of electrons at a pitch angle of $180°$ during the MCL and CIR, and behind those structures, which confirms that STEREO-A is in the towards sector, North of the HCS, at this point. The lack of counter-streaming in the MCL suggests that the magnetic field lines are open, thus only one end of the magnetic field lines which form the flux-rope are rooted at the Sun. A magnetic field reversal with electrons reversing in pitch angle from 0 to $180°$ appears close to the start of the MCL. Thus, the reversal is likely to be a crossing of

the HCS (see Crooker et al., 1993). This can indicate that the MCL is acting as an occlusion in the current sheet.

At OMNI, the normalised magnetic field, shown in panels (c,d) of Fig. 2 turns from North ($B_Z > 0$) to West (-$B_Y > 0$) to South ($B_Z < 0$) in the MCL, again indicating a NWS cloud with left-handed (negative) helicity. The ACE electron pitch angle data in panel (g) of Fig. 5 shows a similar result to STEREO-A, with outward magnetic field and streaming at $0°$ before the MIR event (Away sector) and inward field and streaming at $180°$ during and after the event (Toward sector), indicating that

ACE had crossed the HCS close to the start of the MCL and is likely north of the HCS when the MCL is observed.

Similarly, at STEREO-B, in panels (c,d) of Fig. 4, we observe the normalised magnetic field turn from North ($B_N > 0$) to West ($B_T > 0$) to South ($B_N < 0$) indicating a NWS cloud, a bipolar flux-rope with a low-inclination and left-handed (negative) helicity. The (normalised) $B_R$, component of the magnetic field (panel (b)) is mostly negative during and after the MCL, again indicating that the magnetic field within the MCL is predominantly orientated towards the Sun. We observe

electrons streaming unidirectionally at a pitch angle of $180°$ along field lines (panel (j)), albeit with some spreading, indicating a location predominantly in the towards sector of the Sun, north of the sector boundary. While again the HCS crossing appears to coincide with the start of the MCL, ahead of the MCL there are magnetic field sign changes associated with brief pitch angle reversals, indicating more structure in the field and electron data than at the other spacecraft. A main field reversal, seen first in $B_R$ turning negative, is observed around 18:30UT on 2 July, about 9.5 hours ahead of the MCL, with electrons streaming at

$180°$. From 21:00 UT, the magnetic field turns back and forth while the electrons are counter-streaming. From 3 July, another main reversal is observed with $B_R$ turning back to positive and with electrons streaming at $0°$. Those field reversals suggest multiple folds in the magnetic field.

## 2.3 MIRs

In this section we address the presence of a CIR, observed at all three near-ecliptic spacecraft and how this combines with the

transient presented earlier to become an MIR (e.g. Burlaga et al., 2003; Rouillard et al., 2009b; Farrugia et al., 2011; Rodkin et al., 2018; Shugay et al., 2018). We provide evidence that the MIR is the result of the fast wind interacting with the slow wind in which the flux rope is embedded leading to the flux rope being entrained in the resulting CIR. In particular we comment on





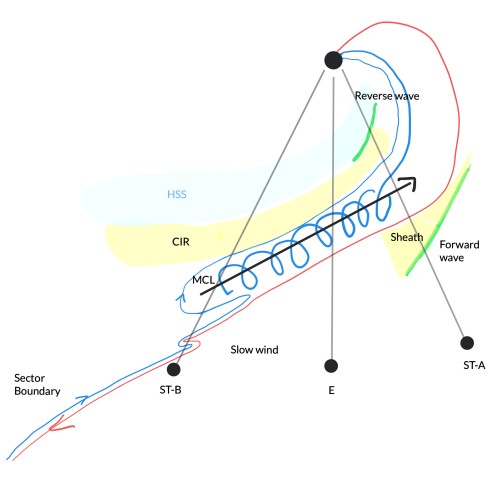

**Figure 6.** Sketch of the MIRs observed at different longitudes near ecliptic, at STEREO-B, Earth and STEREO-A (all within 15 ° longitude of Earth). The view is from the solar north pole. The MIR is shown as the result of the MCL interacting with both the CIR and the HCS. The MCL flux rope with main axis (black arrow) pointing Westward is embedded between the toward Northern sector (blue) and away Southern sector (red) field lines. The MIR variations across the spacecraft demonstrate the progression of the interaction between the CIR and MCL from West to East.

how the arrival times at the three spacecraft are consistent with a flux-rope entrained in a CIR, and on the difference in an MIR observed at different longitudes. Figure 6 shows a sketch illustrating the main features of the MIR.

In the near-ecliptic spacecraft observations, it is clear that, in addition to the presence of a MCL, the in-situ observations resemble a CIR (Pizzo and Gosling, 1994). Despite the lack of forward waves at STEREO-B and OMNI, we are able to match distinct magnetic field rotations and pressure waves across the observations. There is one missing property however. CIRs traditionally show high plasma beta (Borovsky and Denton, 2006). Here we observe a low plasma beta, which is a distinct signature of ICMEs and MC(L)s.

One piece of evidence to indicate the presence and influence of a CIR is given by the arrival timings. This evidence is provided by the observation that the MCL arrives at STEREO-B first, then Earth and STEREO-A, even though STEREO-B is the furthest of the near Earth spacecraft from the Sun. We expect, and in this case observed, timings consistent with a CIR in the solar wind for a MCL that is greatly influenced by its surrounding solar wind. In cases where CMEs are compressed by HSSs we can reasonably expect that the magnetic field lines joining the leading edge of the CME to the Sun are forced to

co-rotate due to the high pressures induced by the compressive effects of the HSS.





These complex interactions offer a probable explanation as to why the event is relatively short, with little compression ahead of the MCL (an ICME front sheath region can only be found at STEREO-A), and why it is also difficult to disentangle the structure in the in-situ data without comparison across different spacecraft. The transients are not reported by any in-situ ICME catalogues relating to the near-ecliptic spacecraft as it is unlikely to have been spotted by usual automated methods and the event duration may be too short to meet catalogue criteria. Kilpua et al. (2012) found that ICME-like transients (in contrast to ICMEs) occur closer to CIRs.

The interaction of CIRs with transients including CMEs is not uncommon. In particular, Rouillard et al. (2008) used HI images to attribute CIR-associated waves to continual release of small-scale transients in the slow solar wind, which are subsequently compressed in CIRs. As seen in Fig. 7, the time–elongation maps, or *J-maps*, from STEREO/SECCHI-HI-A extending form the 19 June to the 5 July 2007 show a nest of converging tracks, which can be interpreted as a pattern of CIR-entrained blobs. We believe the MCL structure is associated with one of those disturbances. One cannot be more specific, but for reference we can take one of the CIR disturbances, $HSIR\_STA\_20070623\_100341$, reported in the HELCATS STEREO SIR/CIR Catalogue (Plotnikov et al., 2016)[1]. The nest of tracks overplotted in Fig. 7 corresponds to a set of blobs all travelling with the same speed as the HELCATS CIR, which is found to have a speed of $302 \pm 21 \, \mathrm{km \, s^{-1}}$ (see Rouillard et al., 2010a, for more details on the uncertainty value). The timings are derived relative to that CIR with the knowledge that one of the blobs therein was fitted to be travelling at $64 \pm 6°$ angular separation from the Sun-spacecraft line. The CIR is predicted to arrive at STEREO-B at 15:15 UT on 3 July and at STEREO-A at 03:13 UT on 4 July 2007. This is similar to the observed earlier arrival time for STEREO-B and separation of ∼12 hours.

Another signature evidencing the MIR are the sector boundaries detected in the magnetic field and the near-ecliptic electron pitch angle data. The data show that the magnetic field is anti-sunward before and sunward during and after the MCL and CIR, which is consistent with the crossing of the HCS near the start of the MCL. Sector boundaries are often found just upstream of CIRs or having just been swept up by the forward waves of CIRs since they are embedded in the slow wind into which the fast wind is running into to create the CIR (Pizzo and Gosling, 1994). The proximity of the sector boundary crossing to the start of the MCL is also consistent with a flux-rope, originating from the streamer belt below the sector boundary as a result of reconnection, locally replacing the HCS, while expanding outwards (Crooker and Intriligator, 1996; Crooker et al., 1998, 2004; Foullon et al., 2011). It is thus likely that this MIR is the result of the MCL interacting with both the CIR and the HCS near the ecliptic.

This type of MIR is not uncommon, and the prevalence of small flux-ropes caught within the leading edge of CIRs observed in situ is discussed by Rouillard et al. (2009b), who confirm that HI observes CIRs in difference images via small-scale transients caught up in the compression region. In fact, Rouillard et al. (2009b) report on a small-scale transient entrained by a CIR that has parallels with our event. Their event is observed later on 20 July 2007. Resemblances between the two events include similar flux-rope profiles observed at STEREO-A with a clear reversal of the polarity of magnetic field lines and a change of suprathermal electron pitch angle from 0° to 180° at the start of the MC(L). For the event presented by Rouillard

---

[1] https://www.helcats-fp7.eu/catalogues/wp5_cat.html





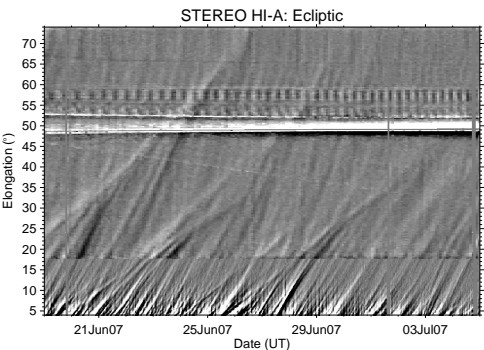 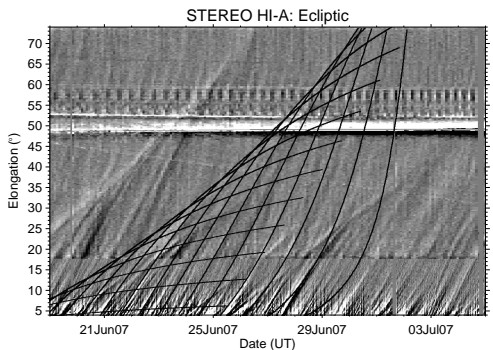

**Figure 7.** STEREO/SECCHI/HI-A *J-maps* for the period from 19 June to 5 July 2007 constructed in the ecliptic at a PA of $280\,^\circ$: (Left) without and (Right) with tracks superimposed. The tracks corresponds to a set of blobs all travelling with the same speed as the blob that corresponds to HELCATS CIR $HSIR\_STA\_20070623\_100341$, which travels at $64^\circ$ angular separation from the Sun-spacecraft line. The trajectories are derived from $180^\circ$ to $0^\circ$ separation angles from the Sun-spacecraft line, in steps of 10 degrees. Each blob is tracked over a somewhat arbitrary time of 10 days.

et al. (2009b), there is no corresponding MC or MCL at the other spacecraft during the same CIR passage, while in our case,
230  we observe clear MCLs at STEREO-B and OMNI, which are smaller and more turbulent than at STEREO-A.

Thus our event allows the comparison of MIR observations at different longitudes. In Fig. 6, the MIR is shown as the result of the MCL interacting with both the CIR and the HCS. From West to East (right to left) the different observations provide in turn snapshots of the extent of the interaction between the CIR and MCL, progressing as we move from STEREO-A to OMNI (Earth) to STEREO-B. In the earlier stage of interaction, at STEREO-A, a forward wave and front sheath are produced ahead
235  of the MCL and a reverse wave is found at the back of the CIR as a result of the interaction with the HSS. In the next stages, at Earth and STEREO-B, the MCLs are smaller and more turbulent and the main sector boundary crossing appears to coincide with the start of the MCL (no sheath). Finally in the later stage at STEREO-B, the MCL appears to be fully entrained in the CIR (noting the higher speed of the MCL at STEREO-B compared to the other spacecraft); ahead of the MCL, small-scale structures in the field and electron data are indicative of multiple magnetic folds, that may result from greater interactions with
240  the HCS (e.g. produced by interchange reconnection with the HCS). Hence, the MIR variations across longitudes demonstrates the radial evolution of the MIR. Due to corotation, the HSS is more caught up with the MCL at STEREO-B and progressivley less so at OMNI and STEREO-A.



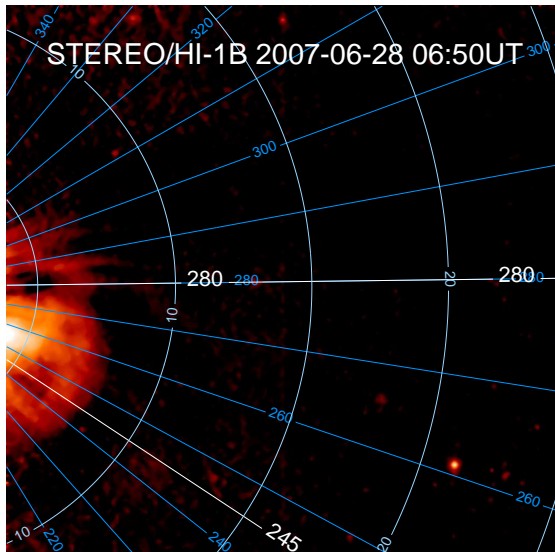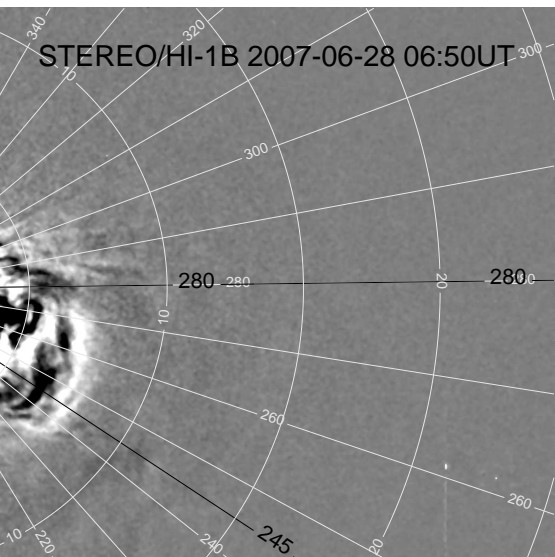

**Figure 8.** HI-1B images of the CME taken at 06:50 UT on 28 June 2007. (Left) Level 2 background-subtracted image provided as part of the standard calibration performed by the PI team (STFC/RAL Space), with additional smoothing applied in order to remove some of the background starfield. (Right) running difference image between consecutive Level 2 images. The position angles at 245° and 280°, over-plotted in both images, represent the two position angles at which the CME disturbances at Ulysses and near-ecliptic were tracked.

## 3 Comparison with *Ulysses*

The origins of the MIR near the HCS as a MCL embedded in a CIR can be the result of reconnection near the sector boundary. During June-July 2007, (I)CME events were relatively rare and isolated as they occurred during a period of solar minimum. Given this and the various ways in which (I)CMEs can change during propagation, we ask the question: is it feasible that the MC observed at *Ulysses* and the MCL near the ecliptic are actually part of the same ejecta or at least have a common origin? In other words, are we observing different substructures of the original CME or associated with the same event?

### 3.1 Self-similar expansion fitting

Here we re-visit the remote-sensing observations presented in Maunder et al. (2022), in which a CME observed in situ as an ICME at *Ulysses* was clearly visible in STEREO/SECCHI/HI-B images. In Maunder et al. (2022), we then applied a self-similar expansion fitting (SSEF) (Davies et al., 2012) to the data points obtained by tracking the CME front in *J-maps* constructed at a fixed position angle (PA) of 245°, close to the CME apex in the STEREO-B HI field of view (see Fig. 8). To estimate the CME's speed and 3D propagation direction, the SSEF works by assuming that the CME front possesses a circular cross-section that expands with a constant half-width (see Davies et al., 2012; Barnes et al., 2020). Maunder et al. (2022) adopted a half width of





**Table 4.** Predicted timings and speeds of in-situ impacts of the CME leading edge determined from SSEF using HI, July 2007.

|  | PA of fit (degrees) | Impact Time (UT) | Speed (km s$^{-1}$) |
| --- | --- | --- | --- |
| STEREO-A | 280 | 05:42 03 July 2007 | 306 |
| OMNI/ACE | 280 | 05:48 04 July 2007 | 270 |
| STEREO-B | 280 | 12:44 05 July 2007 | 242 |
| *Ulysses* | 245 | 12:13 05 July 2007 | 328 |

90 °, equating to the so-called Harmonic Mean geometry (e.g. Lugaz, 2010). This resulted in an apex propagation direction of -28.8 ° latitude, 55.1° longitude [HEE], consistent with its subsequent detection at *Ulysses*.

Using the same STEREO/SECCHI/HI-B images, an additional *J-map* was constructed to track the CME front at a PA of 280 °, which corresponds almost exactly with the Ecliptic at this time (see Fig. 8), giving values of 2.0 ° latitude and 41.3 °
longitude [HEE]. Table 4 presents the speed and, derived therefrom, the predicted impact times of the CME front, calculated at the positions of *Ulysses*, both STEREOs and OMNI/ACE, using the results of both fits. The SSEF method provides CME launch time, propagation direction and speed, which are used to determine impact times at the different spacecraft locations, where we also account for the curvature of the CME front using the method described in Möstl and Davies (2013). The SSEF-derived longitude near the ecliptic implies that the CME is directed 41.3 ° longitude West from OMNI/ACE, thus 31.9 ° from
STEREO-A and 47.7 ° from STEREO-B, indicating that these positions would only be likely impacted by the ICME Eastern flank rather than the nose.

Assuming that the proposed scenario that the CME flank does impact STEREO/ACE is correct, the results, showing that the SSEF speed of the CME apex is larger at the high-latitude of *Ulysses*, are in agreement with the CME apex being faster than its flank. The SSEF method however does not account for complicated morphology and behaviour, particularly CME
interactions with background structures and we note that, contrary to the predicted impact times from SSEF, the event arrives at STEREO-B first (then Earth and STEREO-A). As explored in Sect. 2.3, this is expected as, whilst STEREO-B is the furthest of the near Earth spacecraft from the Sun, the CME is entrained in the CIR near the ecliptic and combines as an MIR. We also note the similarity between the inferred SSEF speeds near ecliptic and the HELCATS CIR speed of $302 \pm 21$ km s$^{-1}$. Thus, the observed timings and inferred SSEF speeds near ecliptic are more consistent with a CIR in the solar wind than the flank of
a CME that is not largely influenced by its surrounding solar wind.

## 3.2  Length scale analysis

The ICME-SIR combination at *Ulysses* may be seen as a MIR, but to distinguish it from the MIR structure near ecliptic, we choose to refer to it as the ICME-SIR. Analysis of the ICME-SIR at *Ulysses* and MIRs near the ecliptic show that they have two to three distinct regions: (1) a front sheath FS (at *Ulysses* and STEREO-A only), (2) a MC(L), and (3) a subsequent
compression region created by interaction between the MC(L) and a HSS. In order to compare these, we calculate the length





**Table 5.** Length scales given with standard deviation for the FS, MC(L), and SIR, their total and the expansion velocity $V_{exp}$ at each of the near-ecliptic spacecraft and at *Ulysses*.

| Spacecraft | FS | | | MC(L) | | | SIR | | | Total | | | $V_{exp}$ |
|---|---|---|---|---|---|---|---|---|---|---|---|---|---|
| | ($10^6$ km) | | ($\approx$ AU) | ($10^6$ km) | | ($\approx$ AU) | ($10^6$ km) | | ($\approx$ AU) | ($10^6$ km) | | ($\approx$ AU) | (km s$^{-1}$) |
| STEREO-A | 10.8 | $\pm$ 0.8 | 0.07 | 17.9 | $\pm$ 0.36 | 0.12 | 26.8 | $\pm$ 4.0 | 0.18 | 55.5 | $\pm$ 12.3 | 0.37 | -7 |
| OMNI | | | 0 | 10.3 | $\pm$ 0.41 | 0.07 | 23.8 | $\pm$ 3.2 | 0.16 | 34.3 | $\pm$ 6.7 | 0.23 | -22.25 |
| STEREO-B | | | 0 | 15.1 | $\pm$ 1.20 | 0.10 | 36.1 | $\pm$ 2.4 | 0.24 | 51.2 | $\pm$ 0.8 | 0.34 | -76.5 |
| *Ulysses* | 6.8 | $\pm$ 0.3 | 0.05 | 32.6 | $\pm$ 1.0 | 0.2 | 12.2 | $\pm$ 1.5 | 0.08 | 51.7 | $\pm$ 4.3 | 0.34 | 14.2 |

scale of each region for each set of observations by taking the mean velocity of the region multiplied by the duration of the region, with the standard deviation included for reference. Durations can be obtained from Table 2 and the estimated length scales for each of the regions at each spacecraft are presented in Table 5.

The MCL durations are relatively short across all observations at 1 AU near the ecliptic, ranging between 0.07-0.12 AU;
there the compression is due to the CIR. This is much smaller than the 0.2-0.3 AU MC average at 1 AU (Lepping et al., 1990; Owens et al., 2005). At *Ulysses* the duration of the MC is 24 hours, the MC has a mean speed of 378 km s$^{-1}$, and an expansion speed of 14.2 km s$^{-1}$. We obtain an estimated length scale of 32.6 $\pm$1.0 $\times 10^6$ km ($\approx$0.2 AU) at *Ulysses*, which is typical for slower ICMEs (Lepping et al., 1990). This length scale is somewhat short at the radial distance of 1.48 AU but expected, since the HSS is propagating directly into the back of the MC, which prevents its trailing edge from decelerating and thus the cloud
from expanding (Maunder et al., 2022).

Additionally, for the MIR or ICME-SIR as a whole, we estimate the length scale of the disturbance to be 0.23-0.37 AU near the ecliptic and $\approx$0.34 AU at *Ulysses*. The interaction regions (CIR and SIR) are approximately twice as large near the ecliptic than at *Ulysses*, indicating different HSS sources originating from the northern sector and the southern sector, respectively. There is significantly more disturbance and a larger front sheath (FS) region at STEREO-A (there is no front sheath at OMNI
and STEREO-B). If the two FS at *Ulysses* and STEREO-A were related, the observations would be consistent with the FS region being thinner towards the nose of the ICME, sampled by *Ulysses*, and thicker at the Eastern flank of the ICME, sampled by STEREO-A, in accord with the expected thickness of the FS increasing from the nose of an ICME (see Kilpua et al., 2017, for a review of ICMEs and their sheath regions).

Assuming no variations in the MC(L) size due to longitude and latitude, a comparison of MC(L) length scales as it travels the
radial distance $R$ of 7.8 $\times 10^7$ km (0.52 AU) between STEREO-A and *Ulysses* indicates that the ICME would have expanded by 14.7 $\times 10^5$ km (82 % increase, see Table 5) over 20 hours 44 minutes, giving an expansion velocity of 20 km s$^{-1}$ as the ICME travels outwards. The non-dimensional expansion rate calculated by dividing the increase in radial extent by $R$ is 0.19. Richardson (2014) noted in his study of 11 ICMEs between 1 AU and *Ulysses* that most ICMEs increased in radial size with $R$ ($10^6$ km), with size variations ranging between R$^{-0.27}$ and R$^{0.91}$. The present case corresponds to R$^{0.62}$, thus within the
same range. However, while Richardson (2014) only considered ICMEs in relative ($\pm30°$) radial alignment, we must exercise





caution in comparisons here as we may be observing different (sub)structures and cannot quantify if this is an effect of the difference in separation angle (combining the large difference in longitude and latitude) rather than of radial expansion.

However, the mean expansion speeds, $V_{exp}$, calculated as half of the difference between the speed of the leading and trailing edge of the near-ecliptic MCL, is negative for all spacecraft, indicative of compression (see Table 5). Such compression is common of transients entrained in CIRs, which is also consistent with the arrival times and the evidence of a CIR-entrained MCL. This MIR is consistent with other findings from MIR studies of entrained flux-ropes. In contrast, at *Ulysses* we have clear signatures of local expansion (declining flow speed and expansion velocity of $14.2 \ \mathrm{km \, s^{-1}}$).

### 3.3 Enlil

In Maunder et al. (2022, see their Fig. 10) we presented simulations of the (I)CME propagation in the heliosphere using Enlil (available via the Community Coordinated Modelling Centre, model run id: Megan_Maunder_062322_SH_2 [2]). This time-dependent 3D MHD model solves equations for plasma mass, momentum and energy density, and magnetic field, with a Flux-Corrected-Transport algorithm to model the ICME (Odstrčil, 2003). The choice of geometric input parameters (see properties in the linked control file[2]) was guided by the STEREO/SECCHI COR2 and HI analysis alongside the observed properties of the MC at *Ulysses*, and the simulation results show the ICME propagating towards *Ulysses*, and a corresponding disturbance seen at *Ulysses* of the same duration but one day earlier than observed.

As shown in Fig. 9, this simulation run also shows the ICME impacting STEREO-A and Earth. At STEREO-A the simulation shows, as seen in Fig. 10, the ICME arriving over one day earlier than observed but approximately the same duration. Using Earth for comparison with OMNI, the simulation shows the ICME arriving around 12 hours earlier and approximately the same duration. We model an event that is very narrow radially but has a relatively large latitudinal and longitudinal extent. Broadly, the model and observation agree on a slow ICME that is launched towards *Ulysses* with its Eastern-flank skimming STEREO -A, -B, and OMNI, supporting one possible interpretation that the observed structures could be related to the same CME or original eruption.

At both STEREO-A and Earth the temperature profiles of the simulated and observed (I)CMEs are consistent (see top right panel in Fig. 10). However,the simulation does not show the rotation in the magnetic field vectors (bottom right panels in Fig. 10). Additionally, as shown in Figs. 9 and 10, the model does not account for the CIR and HCS. The difference between the red (observed at STEREO-A) and blue (simulated) velocity in panel (d) of Fig. 9 is up to $250 \ \mathrm{km \, s^{-1}}$. In this case, the CIR near the ecliptic (in the Northern Sector) succeeding the ICME is not accurately represented. We observe a small increase in solar wind radial flow velocity (green, $\sim 500 \ \mathrm{km \, s^{-1}}$) in panels (a, b) of Fig. 9, but not the observed $600 \ \mathrm{km \, s^{-1}}$ at STEREO-A. Thus the simulation does not account for solar wind speed changes and their interfaces (hence the observed pressure waves are not present in the simulations), nor does it account for sheath and interaction regions that form as a result of shock or waves from propagation and interaction with different ambient wind speeds.

The model inserts a cone CME into the simulated structured background solar wind, which is presumed to be both supersonic and super-alfvenic at the inner radial boundary. These assumptions and potentially small errors in the inner boundary conditions

---

[2]https://ccmc.gsfc.nasa.gov/results/viewrun.php?domain=SH&runnumber=Megan_Maunder_062322_SH_2





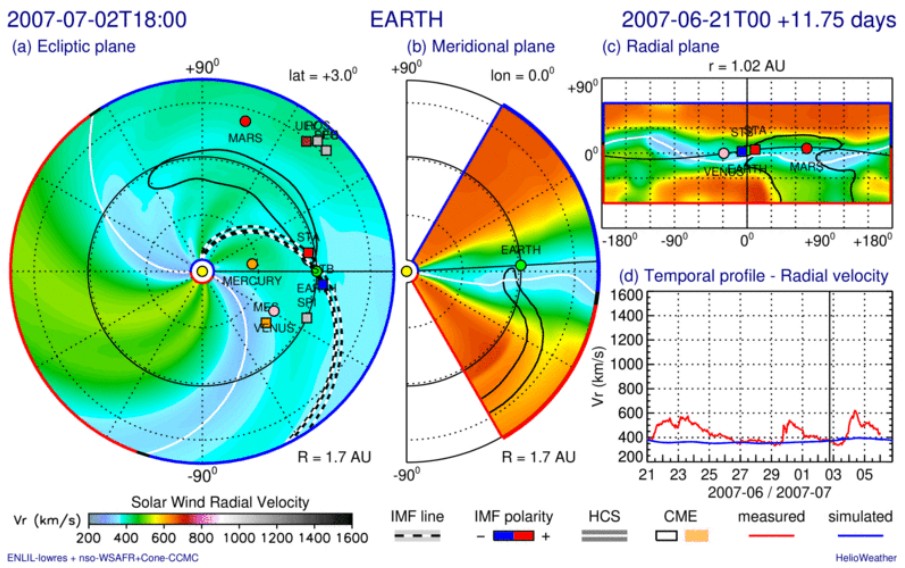

**Figure 9.** A snapshot at 18:00UT on 2 July 2007 of the solar wind radial flow velocity in the heliosphere within 1.02 AU simulated with Enlil in **(a)** the ecliptic plane, **(b)** the meridional plane, **(c)** the radial plane. The simulated CME is shown as a *black contour*. The HCS is shown as a *white line*, the IMF as a *black* and *white dashed line*, and spacecraft and planet positions are projected onto the planes. **(d)** Temporal profiles of the radial solar wind velocity, simulated in *blue* and measured in *red*.

can lead to oversimplifications of the simulation which do not accurately reflect observations. A small latitudinal error in the
position of the current sheet, at the inner boundary, can lead to large errors in modelling in situ profiles.

## 3.4   Global magnetic structure

Based on the modelling and kinetic properties, we expect that *Ulysses* samples the nose of the ICME and the near-centre of the corresponding MC and that the near-ecliptic spacecraft could skim the flank of the ICME. As presented in Sect. 2, the near-ecliptic observations show clear signatures of a MCL which support the flux-rope interpretation; only one end of the magnetic
field lines which form the MCL flux-rope appear to be rooted at the Sun. The lack of compression in the sheath at STEREO-A means that there is insufficient draping around the front of the MCL to create a planar structure and provide a conclusive result using a MVA. However, we deduce a North-West-South (NWS) axis configuration suggesting that the flux-rope has a left-handed (negative) helicity. In Table 3, we compare the flux rope properties with those at *Ulysses*. Examination of the electron pitch angle characteristics at *Ulysses* show that, ahead of the ICME, we observe streaming predominantly away from the Sun
with some evidence of counter-streaming during the event, which is evidence of a closed magnetic topology in the ICME, consistent with the flux-rope interpretation (Maunder et al., 2022). The *Ulysses* analysis was done in the Heliographic RTN system of reference and indicates a West-North-East (WNE) axis configuration (given with approximate corresponding HEE directions as seen from *Ulysses*) suggesting a right-handed (positive) helicity. *Ulysses* has a much higher latitude ($-32.5°$),





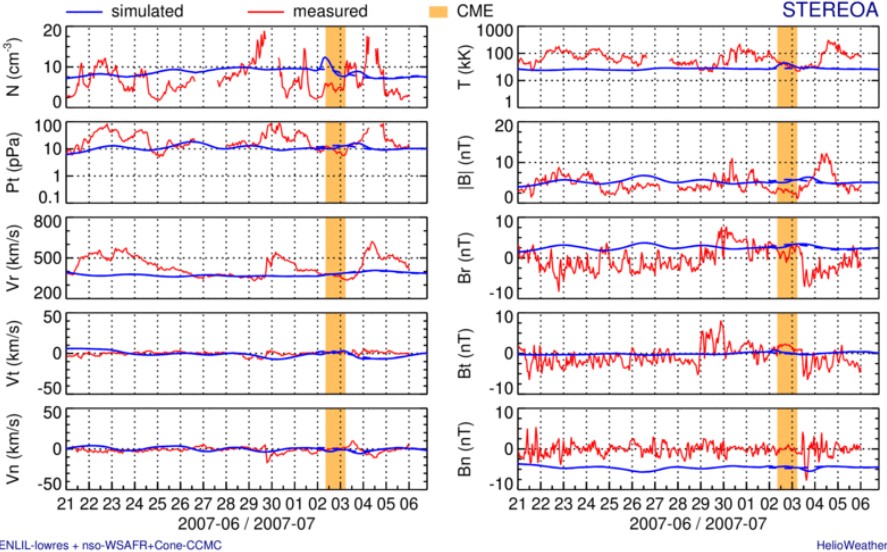

**Figure 10.** A plot of the simulated (*blue*) and measured (*red*) solar wind and magnetic field properties at STEREO-A (left top to bottom then right): density, N (cm$^{-3}$); total pressure, Pt (pPa); radial, Vr, tangential, Vt, and normal, Vn velocity (km s$^{-1}$); temperature, T (kK); magnetic filed strength magnitude, $|B|$ (nT); the radial, Br, tangential, Bt, and normal, Bn, magnetic filed vectors (nT). The *yellow* shade indicates the modelled CME.

than the near-ecliptic spacecraft. The normalised components of the magnetic field are in a system of reference relative to each
spacecraft rather than a global system.

Even if the kinetics are in agreement, the magnetic properties appear to rule out that what is observed near Earth is a subsection of the ICME detected at *Ulysses*. Indeed, conservation of helicity asserts that the flux-rope helicity remains the same through the CME evolution (Ji et al., 1995; Rust, 1997; Li et al., 2011, 2014); Berger and Field (1984) showed that helicity decays in the corona on a diffusion time scale and thus helicity decay cannot be important in CME evolution. Eruptive
events at the Sun associated with CMEs can be used as proxies of CME helicity (Marubashi, 1986; McAllister et al., 2001), as there are no precise observational methods available for estimating three-dimensional coronal magnetic fields, thereby the magnetic structure of CMEs (Pal, 2022). In this case, there is only some coronal dimming visible on the south west of the solar disk, pointing towards an origin in the south hemisphere, which would predict a left-handed (negative) helicity.

However, opposing axis orientations and signs of helicity deduced from the magnetic field structures do not necessarily
contradict the notion that we are observing the same initial structure. Only around 80% of MCs are observed to have the same predicted helicity based on the hemispheric segregation rule (Rust and Kumar, 1994; Bothmer and Schwenn, 1994) and Good et al. (2015) note that it is possible to fit a force-free model flux-rope which has opposite helicity to that deduced by the empirical methods used here. Furthermore, there is a longitudinal separation of $\sim 40\degree$ between *Ulysses* and STEREO-A. In the study by Good et al. (2015) where a similar disparity in handedness was found with a 27 $\degree$ longitudinal separation
(between observations at MESSENGER and STEREO-B, and those at Venus Express), the validity of the model fit at one flux





rope (Venus Express) was questioned. In the study by Rodríguez-García et al. (2022), a large longitudinal difference of 78°
separates the ICME counterparts (in MESSENGER and both STEREO spacecraft in situ observations) of an CME observed
in twin-STEREO spacecraft coronagraph images. A graduated cylindrical shell model was applied to reconstruct the CME
from the coronagraph images and the flux-rope structures detected in situ at all three spacecraft were found to belong to the

same ICME, despite the helicity deduced at STEREO-A and MESSENGER opposing that at STEREO-B. Rodríguez-García
et al. (2022) proposed that this might be due to the fact that STEREO-B intercepted one of the legs of the structure far from the
flux-rope axis whereas STEREO-A and MESSENGER were crossing through the core of the magnetic flux-rope. They referred
to studies that support the possibility of magnetic flux added in opposite directions during the first stages of the CME evolution
(Cho et al., 2013; Vemareddy and Démoulin, 2017), so the centre and the legs of the flux rope can have opposite helicities.

Finally, the radial distance of $7.8 \times 10^7$ km (0.52 AU) between STEREO-A and *Ulysses* corresponds to a time difference of
nearly 21 hours. Studies of ICMEs observed by radially aligned spacecraft are relatively rare and have been used to explore
the structure of ICME flux-ropes in which the stability and low plasma beta of the flux-rope features are preserved in spite
of macro-scale differences in the flux-rope structure (Good et al., 2019). Good et al. (2019) propose that orientation changes
observed by radially aligned spacecraft may be due to flux-ropes aligning with the HCS and the flattening of the ICME front as

it moves out from the Sun or due to the spacecraft sampling different regions of the flux-rope, with the different regions having
different axis orientations. This suggests that global properties of ICMEs such as axis orientation can vary significantly across
small angular distances.

Our study further illustrates how the ambient conditions can significantly affect the expansion and propagation of the CME,
introducing additional irregularities to asymmetric eruptions. We note here that the near-ecliptic spacecraft are close to the

sector boundary at this point. It is thus likely that the MIR is the result of the MCL interacting with both the CIR and the HCS
near the ecliptic. The HSS observed in the towards sector near the ecliptic is distinct from the HSS that is observed at *Ulysses*
in the away sector. The MC and MCL ejectas are influenced by two different types of HSS at different latitudes: a HSS which
produces a CIR near the ecliptic associated to the northern hemisphere (Figs. 2, 3, and 4), and a HSS likely originating from
a south coronal hole at *Ulysses*, producing a SIR in the mid-latitudes (Maunder et al., 2022, see their Fig. 7). Thus the near-

ecliptic MCL is likely to have aligned with the HCS and is entrained in a CIR from the northern sector, while the mid-latitude
flux-rope/MC is interacting with a HSS from the southern sector.

Additionally, both observations and models have shown that a flux-rope may kink and deform during propagation (Manch-
ester et al., 2004), a flux-rope can be eroded (Ruffenach et al., 2015), and that reconnection between a CME front and a
HCS magnetic field in the interplanetary medium can significantly change the overall magnetic topology of ICME flux-ropes

(Winslow et al., 2016). Nieves-Chinchilla et al. (2012) showed evidence for significant reorientation of the flux-rope axis and
Rouillard et al. (2009a) observed the trailing part of an ICME which displayed highly distinct magnetic signatures at different
spacecraft. Near the HCS, flux-ropes are also known to realign and locally replace the HCS (Crooker and Intriligator, 1996;
Crooker et al., 1998; Foullon et al., 2009). Therefore, it is possible that the flux-rope undergoes reconnection, erosion, or even
significant reorientation which may offer some explanation for the different helicities and axis orientations observed. Chen

et al. (2019) discovered a new foot-point drift phenomenon, which shows that the foot-points of erupting magnetic flux-ropes



can drift to a new location during its eruption. This suggests that the CME embedded in the HCS can be connected to one of the different foot-points on either side of the sector boundary, which is used to indicate the helicity of the flux-rope (and its location with respect to the sector boundary). If the two events at *Ulysses* and near ecliptic are related, this offers more explanations as to why a different helicity is observed across large separations in heliospheric latitude, longitude, and radial distance, and

given the considerable influence of the surrounding local solar wind conditions, including the proximity to the HCS near the ecliptic. The electron pitch angle characteristics, flux-rope orientations, and heliospheric orientations are consistent with this.

## 3.5 Distortion

At launch, CMEs are often approximated as locally cylindrical objects with circular cross sections, yet they propagate almost radially away from the Sun into the bulk solar wind, which can severely flatten the initially circular cross section. From MVA

analysis presented in Maunder et al. (2022), we obtained the orientation of the two shocks, that point northwards at *Ulysses*, consistent with a concave, inward curved geometry, also known as pancake geometry (e.g. Riley and Crooker, 2004; Savani et al., 2011). The ICME is clearer at *Ulysses*, which is located closer to the direction of propagation, thus sampling nearer the centre of the flux-rope, and is significantly below the HCS, thus giving a less complex solar wind environment. In contrast, spacecraft near the ecliptic are sampling the flank of an ICME/MIR or ICME substructure and are observing the distortion of

the structure as a result of interaction with the CIR/HCS. The argument that we would expect the structure to be less clear with increasing radial distance would only hold if the spacecraft were in radial alignment, close to the direction of propagation of the ICME; in this case we have a large (latitudinal) separation between spacecraft as well so we expect to be observing different parts of the ICME structure (discussed in Sect. 3.4). Furthermore, we have demonstrated that the rear compression regions observed are caused by different solar wind environments: the pressure wave FW2 observed by STEREO-A is likely

a result of MCL and Toward Sector HSS-CIR interaction whereas at *Ulysses* we observe a shock wave from MC and Away Sector HSS interaction.

We deduce that the ICMEs, related or unrelated, have not only 'pancaked' but have also been distorted by their interaction with the surrounding complex solar wind environment. Similar observations have been reported for the flattening and stretching of CMEs and associated ICMEs, indicating that ICMEs can deviate away from the ideal cylindrical structure with circular

cross section (Savani et al., 2011) and for ICME interactions with solar wind streams, CIRs, and the HCS (e.g. Schmidt and Cargill, 2001; Odstrčil and Pizzo, 1999; Jian et al., 2008). It is likely that this continues as the CME propagates through the heliosphere, demonstrating that the assumption that ICMEs undergo a spherical, self-similar radial expansion is only an approximation for, at least some, ICMEs. This has some relevance for the SSEF method that assumes a spherical, self-similar radial expansion but also does not account for complicated morphology and behaviour, particularly CME interactions with

background structures. More sophisticated HI models that use HI time-elongation profiles to predict CME arrival can assume an elliptical front (ELEvoHI, Rollett et al., 2016).



## 4  Conclusions

In conclusion, using remote-sensing and in-situ observations and across multiple near-ecliptic spacecraft with complimentary methods of analysis, we have investigated a MCL observed in-situ on 3-4 July 2007 near the ecliptic and compared it with
the (I)CME at *Ulysses* studied in Maunder et al. (2022). In Sect. 2, we presented in situ observations and provided evidence that what was detected near the ecliptic was a MCL and MIR. This event allowed the comparison of MIR observations at different longitudes showing differences in size, formation of sheath, presence of forward and reverse waves and small-scale structuring, and providing a sequence of different interaction stages progressing from West to East. In order to explore its origins, we conducted further analyses in Sect. 3, through the application of a SSEF method to STEREO HI data and length
scale analysis to in situ observations and compared the results to the observations at *Ulysses*; we also employed Enlil, a time dependent 3D MHD model, to assist in the interpretation of the (I)CME evolution and contributed to the discussion on whether or not this could be the same event or two related events.

  This problem is fundamentally asymmetric in nature. We have observations of a slow ICME that propagates almost directly towards *Ulysses* but through modelling (from SSEF analysis of HI data and Enlil simulation), we have shown it is possible that
the flanks of this ICME or a related ICME substructure also skimmed the near-ecliptic spacecraft. The ICME/MC has a HSS (from the Southern away sector) propagating directly into the back of it and the Enlil simulation shows a narrow radial width but large longitudinal and latitudinal extent. Near the ecliptic we have an ICME/MCL entrained in a CIR (from the Northern toward sector), to create a MIR. Length-scale analysis appears to be consistent with this configuration and indicates that, if the ecliptic MCL is a flank cross of the ICME observed by *Ulysses*, it is smaller than the MC observed at *Ulysses*. Moreover the
front sheath region is smaller at *Ulysses* than at the eastern flank of the ICME, where the sheath region is present at STEREO-A but is not evident near Earth (OMNI) and STEREO-B. However the modelling methods have caveats and limitations. It may be argued that the Enlil modelling results are dependent on the observational inputs, such as the initial CME width, and in the case of complicated morphology and behaviour, it may be helpful to employ a more sophisticated SSEF method (with an elliptical front). Nevertheless, due to other difficulties in establishing connections described below, the extent to which changing input
parameters or models could yield similar or different results is unlikely to affect our conclusions.

  Indeed, there are three main properties challenging the connection between the near-ecliptic MCL and the MC at *Ulysses*: local expansion velocities, magnetic flux rope axis orientations and helicities. From local expansion velocities of the MCL/MC, we observe compression near the ecliptic and expansion at *Ulysses*; from the magnetic structure of the MC and MCL, we observe different magnetic flux rope axis orientations and helicities. We have presented evidence that the near-ecliptic MCL
has been 're-aligned' by its interaction with the surrounding solar wind environment (CIR) and demonstrated that the local solar wind environment can significantly affect the expansion and propagation of an ICME. Opposing signs of helicity could provide indications of flux added in the first stages of CME evolution or magnetic reconnection with the HCS. We concluded that the differing properties do not necessarily contradict the notion that we are observing the same initial structure. These observations and analyses challenge the existing questions around (I)CME development, propagation and helicity conservation
and poses questions for the further study of ICME-ICME connections and their interactions with complex solar wind.



Regardless of the connection between *Ulysses* and near-ecliptic observations, we have presented a comprehensive analysis and discussion of this event as a challenge for observing ICMEs not in alignment, across different latitudes, and in multiple complex solar wind environments. The use of multi-spacecraft analysis remains key to disentangling MCs/MCLs from their solar wind environments and to determine connections between events observed in the heliosphere. Useful extensions, but

beyond the scope of this study, could be to consider energetic particle effects and interplanetary scintillation observations (e.g. Richardson, 2018, and references therein). Nevertheless, the study demonstrates that, in comparison to the identification of ICMEs and MIRs, the origin and formation of ICME-like transients and ICME substructures are less well understood.

*Data availability.* The data-sets generated during and/or analysed during the current study are available from the corresponding author on reasonable request.

*Author contributions.* All authors contributed to the design and implementation of the research, to the analysis of the results and to the writing of the manuscript.

*Competing interests.* The authors declare that they have no conflicts of interest.

*Acknowledgements.* M.L.M. acknowledges financial support from the UK Science and Technology Facilities Council (STFC) studentship, grant number: 2072927.

The HI instruments on STEREO were developed by a consortium that comprised the Rutherford Appleton Laboratory (UK), the University of Birmingham (UK), Centre Spatial de Liège (CSL, Belgium), and the Naval Research Laboratory (NRL, USA). The STEREO/SECCHI project, of which HI is a part, is an international consortium led by NRL. We recognise the support of the UK Space Agency for funding STEREO/HI operations in the UK.

Data analysis was performed with the AMDA science analysis system provided by the Centre de Données de la Physique des Plasmas

(CDPP) supported by CNRS, CNES, Observatoire de Paris and Université Paul Sabatier, Toulouse. The Enlil Model was developed by D. Odstrčil, currently at George Mason University; simulation results have been provided by the Community Coordinated Modeling Center at Goddard Space Flight Center through their publicly available simulation services (https://ccmc.gsfc.nasa.gov).





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
