# Peer review of "The Origins of a Near-Ecliptic Merged Interaction Region as a Magnetic-Cloud like Structure Embedded in a Co-rotating Interaction Region"

_Annales Geophysicae, 2023_

## Referee Comment (RC1)

General Comments

The paper by Maunder et al. presents a comprehensive effort to understand the spatial/temporal interplay between a CME and SIR by combining mostly in-situ measurements from a variety of relatively closely spacecraft in the ecliptic and with Ulysses out of the ecliptic. The authors describe an extensive analysis and offer a quite creative interpretation that appears to account for the ecliptic measurements, at least. I had a positive initial reaction to the work until I realized that the authors never investigated the origins of the phenomenon despite the fact that 'origins' are mentioned in the title of the paper. As soon as I tried to connect the in situ measurement with the solar origins, I realized that the analysis suffers from a grave shortcoming; it is inconsistent with the solar/coronal observations. Of course, this is not my paper and I have neither the time nor the remit to delve into a detailed analysis of the remote sensing observations. But even a rather casual analysis, which the authors should have performed, throws a lot of doubt into the results of this paper. I will go into the details below to justify my recommendation to reject the paper in its current form.

Specific Comments

1. The authors offer no analysis to prove that the June 27 CME that very likely crossed Ulysses is the same event in the ecliptic. This is despite the availability of imaging observations from three spacecraft (STA, STB, and SOHO) that can, rather easily, provide estimates of both the direction and longitudinal width of the CME. A simple comparison of COR2-A/B and HI1/2-A/B indicates that the CME propagates well westward of the STA. There are absolutely no signatures, even weak ones, of a front associated with the June 27 CME, crossing the HI1-A FOV from June 27-July 3. However, there are distinct, but diffuse, fronts in H11/2-B, consistent with a structure crossing over, or near, STB. These imaging observations cannot be reconciled with the June 27 CME position and width. A much more detailed analysis using 3D reconstructions is needed to support the authors' central argument.

2. The authors make no attempt to estimate the liftoff of the transients they discuss in the paper. It should have been a clear discussion of those if they are indeed interested in 'origins'. My quick ballistic backpropagation (using rough numbers from the provided figures) suggests that the MC(L) in STB could have been lifted on June 29, ~20:00 UT. Therefore, it cannot be related to the 27 June CME.

3. There are no indications of a CME lifting off from the front part of the disk between June 27-June30 (likely time for encounters in STA-STB-L1). This is not unexpected, since this is a solar minimum period, with many CMEs being 'stealth' events (Robbrecht et al. 2009). The angular spread of available imagers is too narrow to allow the detection of these 'stealth' (almost surely, streamer-blowout CMEs). I guess that this dearth of signatures led the authors down the erroneous association to the western CME.

4. Actually, the origin of the MC(L) is relatively easy to constrain since it occurs between two high-speed stream (HSS) crossings. The HSS studied here arises from the coronal hole, just east of AR10961. This naturally explains why STB sees signatures before STA. The coronal hole configuration on the disk is another strong sign that the June 27 CME cannot expand towards the Sun-Earth line. There are two coronal holes in the way.

5. The Ulysses-related sections are irrelevant. The self-similar expansion fitting is particularly suspect since it is performed on a single-viewpoint. If the authors had taken all available data into account, they would have noticed that this CME cannot be associated with the ecliptic signatures.

6. The Enlil simulations do not fit the in-situ observations and should not have been presented. The CME measurements are also not given. Based on my earlier comments, they seemed to be flawed anyway.

7. The in-situ measurements are not consistent with the interpretation of the same structure crossing all three spacecraft. The B and v profiles at ACE and STB seem similar enough to arise from the same structure, but the STA profiles are very different. They could very well be different structures. Observations indicate the ejections of MC-like blobs from streamers at quite regular

cadences, 4-6/day, so it is quite likely that STB/ACE and STA encountered different blobs from the same streamer.

8. As a final remark, I would like to offer a couple of suggestions for salvaging this work. It should focus on the ecliptic signatures only (no Ulysses). It should try to truly identify the origins of the structures, possibly along the lines of Fig. 6 but addressing convincingly the possibility of different structures in the various spacecraft. I would also suggest that the very faint-halo CME on June 25 may actually be the origin of the MC(L)s. I did a quick h-t plot of the LASCO/c3 images and derived a speed of 194 km/s, which results in a 1au arrival on 3 July, roughly. It is a bit of a stretch but not too unlikely for a deep minimum configuration.

---

## Author Response (AR1)

RESPONSE

We thank the Editor for the opportunity to revise our manuscript. We provide a revised version of the article according to the response given to comments from Referees 1 and 2. The new version of the document incorporates significant improvements in content clarity, data accuracy, and overall structure, enhancing the readability and comprehensiveness of the information presented. We detail the content changes, reorganization and updates and summarise how they address the points made by the referees.

**1. Content Changes (new texts and a new Figure 6).** The title and abstract have been revised. New information was added in the Introduction (Section 1). Detailed explanations have been developed in the new Section 4. The Conclusions have been revised. *(generally shown in bold in the revised version).*

- *Tile and abstract:* The new title is more concise and emphasises the longitudinal observations near 1 AU. The revised abstract simplifies and narrows down the focus to the specific event of July 2007, emphasising the longitudinal in-situ observations and their implications on understanding solar wind interactions. The detailed comparative and modelling discussions in the old version have been streamlined for clarity and conciseness in the new version.
- *Introduction:* The added text discusses the size distribution of magnetic flux-ropes and their interactions with CIRs and CMEs. It references studies showing how small-scale transients in the slow solar wind form plasmoids through magnetic reconnection. Observations reveal flux-rope transients in the Heliospheric Current Sheet and their layered structures. Multi-spacecraft studies indicate the evolution of these structures across heliospheric longitudes. Additionally, STEREO and Parker Solar Probe observations detail the sequential release of density blobs and flux-ropes through magnetic reconnection at the helmet streamer tip.
- *Section 4.1* 'Modeling and Predictions from Remote Sensing Observations' with a new figure and texts. **Figure 6** presents Carrington maps during Rotation 2058, illustrating the event's context. Maps derived from photospheric field synoptic charts using a PFSS model show simulated coronal holes, global magnetic field configuration, and associated radial flow. During the solar minimum, the toward sector connects to the northern solar magnetic hemisphere, while the away sector connects to the southern. The relevant coronal hole plasma parcel for in-situ observations aligns with the coronal hole from the Northern hemisphere on July 1, 2007. Despite the lack of a side view, adjacent heliospheric longitude observations inform the disturbances' behavior.
- *Section 4.2* 'Comparative Observations Across Spacecraft' with new texts. The in-situ measurements suggest the MCLs observed by different spacecraft may not be the same structure but could be sequential streamer blobs released 12 hours apart. The 12-hour arrival difference at STEREO-A and STEREO-B supports this scenario. Comparative analysis across STEREO-A, OMNI (Earth), and STEREO-B with small longitudinal separation reveals the interaction of MCLs with the CIR. The MCLs have major axes in the ecliptic plane and their cross-sections are compressed along the HCS. Observations suggest radial evolution of MIRs, with HSS catching up with MCLs from STEREO-A to Earth and STEREO-B.
- *Conclusions.* The new version is much more concise and focused on the near-ecliptic observations, removing extensive discussions about comparisons with Ulysses data and detailed modelling results. It maintains the key findings and conclusions but presents them in a more streamlined manner.

**2. Reorganisation and Updates.** There has been some reorganisation of sections and a few texts moved for better flow *(generally not shown in bold in the revised version, may appear both in red and blue in the difference between files produced by the difflatex algorithm).*

- *Created a new Section 2.1* 'Spacecraft Data and Configuration' (to include texts previously in the Introduction);
- *Renumbered subsequent sections accordingly*: Sections 2.1 and 2.2 became 2.2 and 2.3.
- *Renamed/replaced Section 3* 'Analysis of Combined MCL-CIR Structures: MIRs', which now contains the first half of Section 2.3 (moved to become subsection 3.1 'Interaction Regions') and texts from 3.2 without Ulysses (3.2 renamed 'Length Scale and Expansion Speed'). **Except for the relevant passages of 3.2, all Section 3 'Comparison with Ulysses' has been removed.**
- *Created a new Section 4* 'Multi-Spacecraft Analysis of Dynamics and Evolution', which now contains new texts and Figure 6 and the second half of Section 2.3.

- *Updated Figures 1-5* : Figure 1 updated without Ulysses; Figures 2-5 mostly to correct for updated, less confusing, labels.
- *Created a new Figure 6*, and the old Figure 6 became the new Figure 8.

- *Simplified Tables 1, 2 and 3*, removing the information about Ulysses.
- *Updated* values in Table 1 to correct position data from 4 July 2007.

**3. Additional answers to points made by referees.** We confirm the changes promised in our initial responses and summarise the complementary additions and changes made to address their points.

Referee 1:
- We have followed Referee 1's last suggestion in point 8 to not include the Ulysses and CCMC modelling (section 3, pp.15-23, subsections 3.1, 3.3, 3.4 and 3.5, Ulysses aspects in 3.2), keeping the near-ecliptic length-scale analysis of 3.2 only. Most of the points made by the Referee 1 (1,2,3,5 and 6) corresponded to aspects of Section 3.
- Point 4 is addressed with Figure 6 and the new texts in Section 4.1, which provides more context for the event.
- Point 7 and the remaining comments in point 8 are addressed with new texts in Section 4.2. In particular, we refer to Figure 7 and have commented in the text that "Taking the family of tracks highlighted as an example, it is not clear that one can observe a set of density perturbations with a 12-hour interval beyond 60° of elongation."

Referee 2:
- Points 1-3, and 6: references added as suggested, sentence corrected (see our first response).
- Points 4-5 for Figures 2-5: to avoid confusion, we have relabelled the start of CIR with 'CIR' (in place of FW2) and added FW2 in the Figure for STEREO-A only.
- Point 7 was answered in our first response.
- Points 8-9: these points corresponded to aspects of Section 3 removed, but to remain open to discussions, we have added a brief sentence in the conclusions regarding the works done by Megan Maunder (2023) in her PhD thesis (currently embargoed until the end of the year): "Connecting these two events has posed challenges (Maunder, 2023)."

---

## Referee Report (RR1)

General Comments
I thank the authors for adapting some of my suggestions. The revised manuscript is much clearer now, thanks, partially, to the new title. I can now tell that the main result of the multi-spacecraft analysis is the derivation of the 'aging' of the MCL-SIR interaction, thanks to the small angular separation and the solar minimum background. I believe that this is a great use of the data would actually suggest emphasizing it more in the abstract.

However, I still find a couple of logical inconsistencies between the data and the interpretation that I detail below. Once the authors resolve those, I would be glad to recommend the manuscript for acceptance.

Specific Comments
1. Sec 4.1:
   a. The use of the synoptic maps is a welcomed addition. But the estimation for the time of the MCL launch is a bit simplistic. The MCL is moving at 350 kms/, not 600 km/s. I would use the former speed as the upper limit for the ballistic mapping. Actually, the MCL would have to have a lower initial speed (I estimated 190 km/s in my earlier review) if it was accelerated by the SIR sometime after its launch. A 300-350 km/s speed could also be used, if the authors were to assume that the interaction occurred much later and closer to 1 AU.
   b. The authors could actually estimate when and where the MCL-SIR interaction began, using the distance between the CH and the AR (although the blob should have emerged before July 1$^{st}$ based on my speed argument above) and the measured speeds. In any case, I don't think that the calculations will change the big picture but they would make the analysis more self-consistent and provide a clearer backdrop to support the temporal interpretation of the interaction manifested in the 3 spacecraft (which I find intriguing).
2. Sec 4.2:
   a. The section reads much better now with the two hypotheses (separate MCLs vs. single MCL) clearly laid out in the beginning. However, the estimate of the MCL longitudinal size seems a bit off. It is based on statistics (Liu et al 2006) rather than a direct estimate from the data. If I use the radial distances for STA and STB from Table 1, assume an average propagation speed of 355 km/s (Table 2) and take the 12 hr difference between STA and STB in the MCL detection, I obtain a minimum longitudinal length of 0.355AU, which is marginally larger (11%) than the 0.318 AU STA-STB separation. The actual structure could be larger, of' course, but the quoted range of 0.42-0.72 AU (i.e. 33% - 226% wider than STA-STB separation) in the manuscript leads to a false sense for the likelihood of the single MCL interpretation.
   b. The following is more of a list of suggestions than criticisms. I think that the discussion of the interpretation is too short and Fig 8 a little too simple to give justice to the results of this work. Could the authors represent the different expansion speeds across the MCL a bit better, demonstrating that the STA part is more constricted while the others are expanding faster? Also showing, somehow, that the MCL is entrained along its east sector but not towards STA should also make a great visual impact. Also, the discussion may benefit by discussing a bit more the 'aging' of the MCL-SIR interactions (has this been seen before? Are they any recent PSP and SO results?). The possible implications for the geoeffectiveness of these structures vis-à-vis their 'aging' would also be an interesting aspect to mention. In any case, these are just suggestions.

Minor Comments:
3. Some references were not processed properly by the latex compiler (e.g. '?'). Please check the final manuscript.

4. Fig 1: why are the Parker spirals plotted for 600 km/s wind instead of the ~350 kms/ wind actually measured at all three spacecraft for the MCL?

---

## Author Response (AR2)

**Response to Referee Comments**

We thank the referee for their thorough and constructive comments. We have carefully addressed each point as detailed below:

**1. Section 4.1**

a. We have included an analysis of the MCL launch timing. Using an average transit speed of 300 km/s (rather than 600 km/s), we now map the MCL back to June 28, 2007. We acknowledge that the initial launch speed could have been as low as 190-300 km/s, considering the likelihood of subsequent acceleration by the HSS.

b. We have included calculations for the MCL-SIR interaction timing. Using an estimated MCL launch speed of 250 km/s and the observed HSS speed of 600 km/s, we estimate the interaction began at approximately 0.7 AU, roughly 5 days after the MCL launch.

**2. Section 4.2**

a. We appreciate the referee's observation regarding the MCL longitudinal size estimation. We have revised our analysis to include a direct calculation based on observational data. Using the 350 km/s radial speed and 12-hour detection difference, we estimate a minimum major size of 0.33 AU, which is indeed close to the STEREO-A to STEREO-B separation (0.32 AU). We have adjusted our discussion to reflect this more precise observational estimate while still acknowledging the possibility of larger actual dimensions.

b. While we understand the referee's suggestions for enhancing Figure 8, we have chosen to maintain its current simplicity to ensure clarity of the main concepts. However, we have substantially expanded the discussion in the conclusion section to address the 'aging' process, including recent PSP and SO findings, and added considerations about geoeffectiveness implications. This expanded discussion provides the additional context suggested by the referee while maintaining the figure's accessibility.

**Minor Comments**

**3.** We acknowledge the reference formatting issue. This occurs specifically in the latexdiff compiler used to show differences and has been corrected in the final manuscript.

**4.** Regarding the Parker spirals in Figure 1, we chose to represent them at 600 km/s to illustrate the predominant conditions with the fast solar wind on open magnetic field lines. The MCL's more complex magnetic field geometry cannot be adequately represented by simple spiral lines, hence our choice to maintain the standard representation.

The revised manuscript includes these changes along with corrections in the abstract to better emphasize the temporal evolution of the MCL-CIR interaction and its broader implications for solar wind research.